# Identification of a sub-population of synovial mesenchymal stem cells with enhanced treatment efficacy in a rat model of osteoarthritis

Nedaa Al-Jezani[1], Asmaa Affan[1,2], Catherine Leonard[1], Nabangshu Das[1], Luiz Gustavo Almeida[1], Daniel Young[1], Anand O Masson[1,2], Antoine Dufour[1,3], Paul Salo[1,4], Pam Railton[4], James N Powell[1,4], Roman J Krawetz[1,2,4,5]*

[1]McCaig Institute for Bone and Joint Health, University of Calgary, Calgary, Canada; [2]University of Calgary, Biomedical Engineering Graduate Program, Calgary, Canada; [3]Department of Physiology and Pharmacology, University of Calgary, Calgary, Canada; [4]Department of Surgery, University of Calgary, Calgary, Canada; [5]Department of Anatomy and Cell Biology, University of Calgary, Calgary, Canada

*For correspondence:
rkrawetz@ucalgary.ca

## eLife Assessment

This **important** study characterized and identified clonal MSC populations from human synovium. The authors provide **convincing** evidence that clonal MSC populations can be isolated and expanded from both normal and osteoarthritic synovium and that CD47 represents a potential marker for improved chondrogenic potential of MSC sub-populations. These findings could provide new avenues for osteoarthritis treatment in the future and deeper mechanistic understanding of the factors involved in the repair.

**Abstract** Osteoarthritis (OA) is a painful, debilitating disease with no cure or treatments that can predictably stop/reverse its progression. Treatment is particularly difficult since articular cartilage lacks intrinsic repair capacity, despite mesenchymal stem cells (MSCs) being present in the joint with robust chondrogenic potential. While heterogeneity exists among MSC subtypes within human synovium, it remains unclear which populations can regenerate cartilage or impact OA progression. We clonally isolated MSCs from normal and OA patient synovium using indexed flow cytometry, then characterized them through differentiation assays and quantitative proteomics. MSC clones were transplanted into a xenograft rat OA model and evaluated by histology and immunofluorescence. We identified heterogeneity in putative MSCs within and between patient groups and their repair capacity in the rat model. However, traditional cell surface markers could not distinguish these subtypes, highlighting the need for single-cell level understanding. Using unbiased proteomics, we identified CD47 as a novel MSC marker. CD47$^{Hi}$ cells demonstrated robust treatment efficacy in the rat OA model and directly contributed to new articular cartilage formation. Characterizing MSC subtypes is essential for identifying candidates appropriate for clinical investigation and exploiting functional MSCs for cartilage regeneration strategies.

## Introduction

Osteoarthritis (OA) is a chronic joint disease that is characterized by progressive degeneration of the articular cartilage and results in structural changes throughout the joint, including deformity in the subchondral bone, loss of muscle tissue, and inflammation in the synovial membrane (*Arden et al., 2021*; *Grandi and Bhutani, 2020*; *Ni et al., 2020*). These alterations to joint tissues ultimately lead to disability, which in turn results in a major socio-economic burden for impacted individuals and their respective healthcare systems (*Fellows et al., 2016*; *Palazzo et al., 2014*; *Salmon et al., 2016*). Developing a therapy for OA is particularly challenging for multiple reasons, including the multifactorial and elusive nature of the disease (*Hafsi et al., 2019*; *Mora et al., 2018*); and inherent properties of the cartilage (absence of blood vessels and nerves), which results in limited endogenous repair capacity (*Fellows et al., 2016*; *Fernandes et al., 2018*). All current approved treatments for OA patients focus on symptoms, such as pain management but fail to address the core issue of OA; the degeneration of the cartilage and related structural changes within the joint. In view of these limitations in treatment options, researchers have extensively investigated the use of MSCs (also referred to as mesenchymal progenitor/stromal cells) (*Dimarino et al., 2013*) as an alternative treatment modality since they are hypothesized to reduce pain and inflammation through immunomodulatory abilities (*Lee et al., 2015*), while also potentially being able to induce the repair/regeneration of cartilage through direct differentiation into chondrocytes (*Mak et al., 2016*) and/or the release of trophic factors to stimulate indirect repair (*Soland et al., 2013*). MSCs have garnered significant attention over the past two decades, not only because these self-renewing cells possess multi-lineage differentiation capacity but also because these cells have the ability to influence their microenvironment through multiple means (*de Witte et al., 2018*; *Kim et al., 2018*). MSCs have been identified within nearly every mesodermally derived tissue, yet synovial-derived MSCs have been shown to have superior chondrogenic and self-renewing capacity when compared to MSCs derived from a variety of tissues, such as bone marrow and adipose tissue (*De Bari et al., 2001*; *Futami et al., 2012*; *Nishimura et al., 1999*; *Sakaguchi et al., 2005*). Although MSC-based therapies in numerous pre-clinical studies have shown promising results for the treatment of cartilage injury and/or OA (*Jia et al., 2018*; *Krawetz et al., 2023*; *Mak et al., 2016*; *Satué et al., 2019*), outcomes from clinical trials have largely been disappointing, with some studies demonstrating that MSC therapy offers little to no advantage over conventional surgical treatments, such as microfracture or corticosteroid injection (*Koh et al., 2016*; *Mautner et al., 2023*).

In the vast majority of these pre-clinical and clinical studies, cells are isolated and defined as MSCs based on the guidelines proposed by the International Society for Cellular Therapy (ISCT) (*Dominici et al., 2006*). The minimum criteria that must be met include adherence to plastic cell culture-ware, multipotent differentiation capacity (typically bone, cartilage, and fat), and the expression (and/or lack of expression) of a panel of cell surface markers. Unfortunately, even when satisfying these criteria, MSC phenotypic heterogeneity has been observed in animal and human model systems across multiple tissue types (*Phinney, 2012*; *Russell et al., 2010*; *Zhou et al., 2019*). In a previous study, we found significant MSC heterogeneity in clonal populations of cells derived from hip synovium even when cell populations expressed similar combinations of cell surface markers (*Affan et al., 2019*). Therefore, there is a concern that this heterogeneity might negatively impact the capacity of MSCs to demonstrate clinical efficacy both within cohorts and across studies if we can't be sure the same cell type (or subtype) is being isolated and delivered back to patients. Furthermore, since most of the published human MSC data comes from cultured cells, it is also possible that a strong selection bias is induced by the in vitro conditions which modify the properties and populations (subtypes) of the cultured MSCs. Emerging research supports the concept that DPP4$^+$ (also marked by PI16 and CD34) synovial cells function as multipotent stromal progenitors. In postnatal mice, DPP4$^+$ mesenchymal progenitor cells (MPCs) give rise to Prg4$^+$ synovial lining fibroblasts (SLFs) and adipocytes, contributing to normal joint development and synovial homeostasis, with activation and expansion after OA injury. Single-cell transcriptomic atlases further identify PI16$^+$ fibroblasts in perivascular niches as latent reservoirs capable of differentiating into specialized synovial lineages following injury. Together, findings across human and mouse OA (*Collins et al., 2023*; *Knights et al., 2023*; *Li et al., 2024*; *Peters et al., 2025*; *Tang et al., 2024*) models underscore the progenitor role of these universal fibroblasts and highlight the need for a better understanding of the subtypes of MSCs within the joint environment (by identification of the markers that correlate with functional properties advantageous for therapy) might allow for the isolation of specific populations which are best suited for cartilage regeneration applications.

Therefore, this study employed a unique approach to examine the cell phenotype of clonally derived synovial MSCs obtained directly from synovial tissue (without an initial cell culture step). We have: (1) characterized the cell surface marker expression of clonal populations in situ vs. in vitro to determine the robustness of ISCT-recommended markers; (2) correlated the marker expression with multipotent differentiation capacity; (3) determined if the cell surface marker expression could predict MSC differentiation potential; (4) examined the ability of different clonal populations to treat OA in a preclinical rat model; and (5) characterized the global proteome expression pattern of MSCs isolated from normal vs. OA joints and also MSCs vs. non-MSCs derived sub-populations.

## Results

### Clonal MSC derivation

In this study, synovial biopsies were collected from 15 patients undergoing knee orthopedic procedures and 18 normal cadaveric donors (*Supplementary file 1*). From the normal synovium, 228 clonal cell lines were generated, while 259 clonal lines were derived from osteoarthritic (OA) synovium (*Supplementary file 2*). Out of these, 29 clones from the normal synovium and 37 from the OA synovium exhibited the self-renewal potential required to expand to a population size suitable for further analysis, such as differentiation and flow cytometry (*Supplementary file 3*).

To complete all differentiation and flow cytometry analyses (including replicates), each clonally derived synovial MSC line needed to undergo approximately 19 population doublings. However, most clonal cell lines lacked the self-renewal capacity necessary to meet the study's characterization criteria (*Supplementary file 3*).

A negative correlation was observed between donor age and the population doublings of clones derived from that donor (*Figure 1—figure supplement 1*). This relationship was more pronounced in OA patients compared to normal donors. No correlation was found between sex and population doublings (*Figure 1—figure supplement 1*).

### Cell surface marker expression in situ vs. in vitro

The data presented are a representative example from one normal and one OA patient (*Figure 1—figure supplement 2*), while the collective results are summarized in *Supplementary files 4 and 5*. Cell surface marker profiles were compared in situ (pre-culture, indexed cell sorting was used to record cell surface profiles) and in vitro (post-culture). Flow cytometry analysis revealed that cell surface expression of the cloned cells (isolated from both OA and Normal) became altered by exposure to the cultural microenvironment (*Figure 1—figure supplement 2A, B*). Specifically, four clones were obtained from a single normal individual. Clone 1 was initially positive for CD90, CD44, and CD73 in situ and negative for CD105 and CD271, yet this clone gained the expression of CD105 in vitro. Clones 2, 3, and 4 shared similar expression in situ (expressing CD44 and CD73 while lacking the expression of CD90, CD105, and CD271). Similar to clone 1, these clones acquired the expression of CD105 as well as CD90 (*Figure 1—figure supplement 2A*). Clones obtained from a single OA individual show a similar pattern. Clones 1 and 2 were positive for CD44 and negative for CD90, CD73, CD105, and CD271 in situ and gained the expression of CD90, CD73, and CD105 in vitro. Clone 3 was CD44 and CD73 positive in situ and gained expression of CD90 and CD105 in vitro. Clone 4 expressed CD90 and CD73 in situ and gained expression of CD44 and CD105 in vitro.

Collectively, the common trend observed in the clones presented and those from the entire cohort was that despite a heterogeneous cell surface marker expression profile in situ, the marker profile aligned in vitro with nearly all clones (normal and OA) expressing CD90, CD44, CD73, and CD105 (*Figure 1—figure supplement 2*, *Supplementary files 4 and 5*). Since this experiment was performed clonally, we can conclusively state that the individual cells self-regulated their unique cell surface profiles to a common profile once expanded in culture. Overall, this demonstrates a convergent selection bias towards the traditional cell surface marker profile expressed by MSCs.

### Multipotential differentiation assessment

Aside from cell surface marker expression, a hallmark of 'stemness' includes self-renewal ability and multipotent differentiation capacity; therefore, clones were identified as MSCs if they demonstrated differentiation into bone, cartilage, and fat in vitro. The data presented is a representative

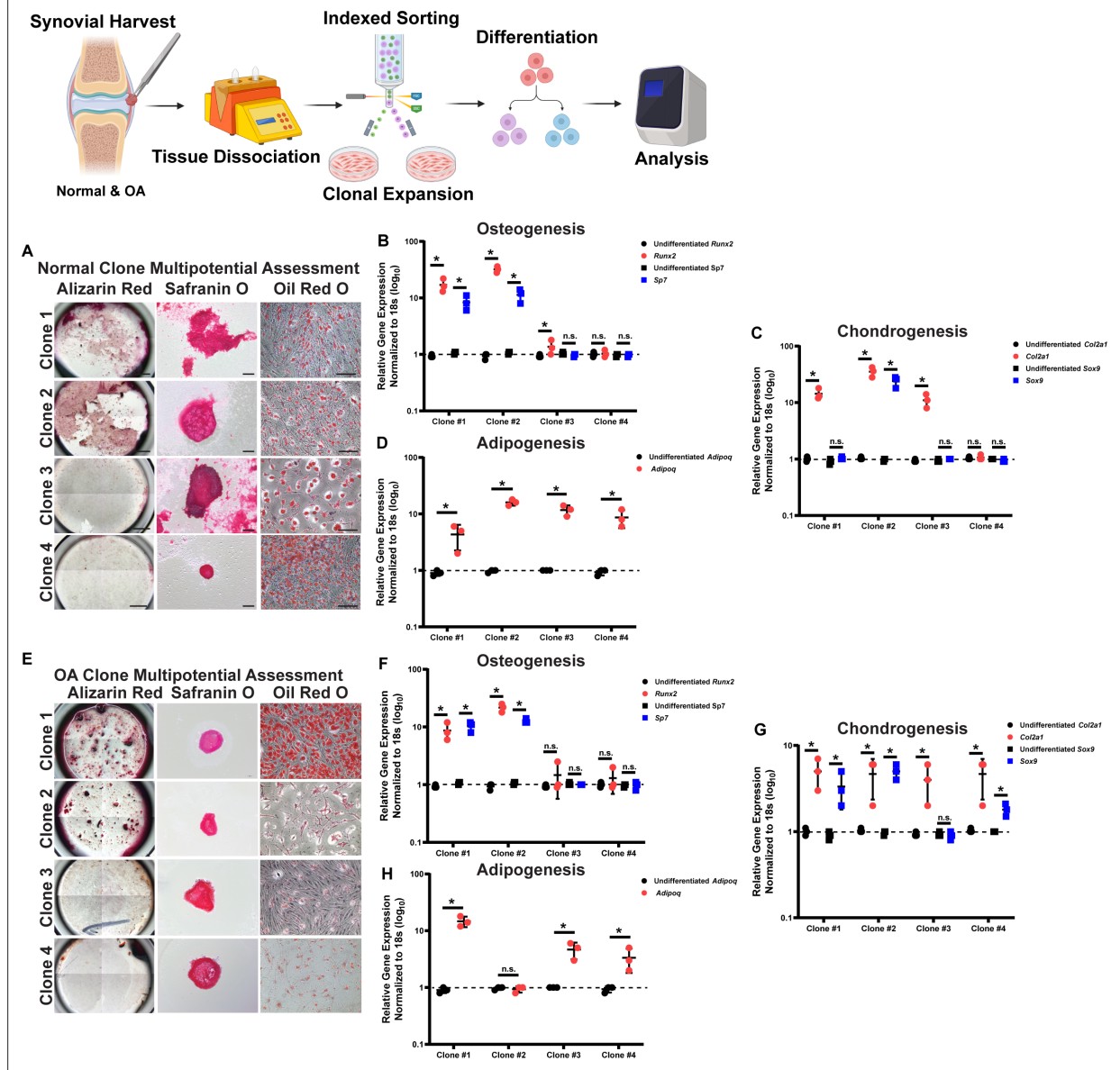

**Figure 1.** In vitro differentiation potential of clones derived from normal individuals and patients with osteoarthritis (OA). Representative histological staining of Alizarin Red, Safranin O, and Oil Red O from four normal (**A**) and four OA clones (**E**). Quantitative PCR (QPCR) data (n=3 biological replicates and n=3 technical replicates, each point is the mean of the technical replicates) is presented for each of the clones for osteogenic (**B, F**), chondrogenic (**C, G**), and adipogenic (**D, H**) markers. Signifiance determined by t-Test. *$p<0.05$, **$p<0.01$. Scale bars equal 200 μm.

The online version of this article includes the following figure supplement(s) for figure 1:

**Figure supplement 1.** Association between population doublings, sex, and age.

**Figure supplement 2.** In situ and in vitro cell surface protein expression on clones derived from normal individuals and patients with osteoarthritis (OA).

**Figure supplement 3.** Overview of the experimental design employed in the current study.

example from <u>one normal</u> and <u>one OA</u> patient (*Figure 1*), while the collective results are summarized in *Supplementary files 4 and 5*. Based on histological analysis, normal clones 1 and 2 stained positive for Alizarin Red, Safranin O, and Oil Red O when induced to differentiate into osteoblasts, chondrocytes, and adipocytes (*Figure 1A*). When qPCR analysis was performed, clone 1 expressed *Sp7* (osteogenic), *Col2A1*, and *Sox9* (chondrogenic), but lacked the expression of *Adiponectin* (adipogenic) (*Figure 1B–C*). Clone 2 upregulated all differentiation markers (*Figure 1B–C*). Clones 3 and 4 lacked Alizarin Red staining but stained positive for both Safranin O and Oil Red (*Figure 1A*). Clone

3 expressed *Sp7*, *Col2a,* and *Adiponectin* (*Figure 1B–C*) while clone 4 only expressed *Adiponectin* (*Figure 1B–C*).

Similar patterns were observed in the clones derived from OA patients. Clone 1, when induced to osteogenesis, chondrogenesis, and adipogenesis, was positive for Alizarin Red, Safranin O, and Oil Red (*Figure 1E*) and expressed *Sp7*, *Runx2*, *Col2a1,* and *Adiponectin* (*Figure 1F–G*). Clone 2 stained positive for Alizarin Red, Safranin O, Oil Red O, and expressed osteogenic and chondrogenic markers but lacked the expression of *Adiponectin* marker (*Figure 1F–G*). Clones 3 and 4 were negative for Alizarin Red but stained positive for Safranin O and Oil Red O (*Figure 1E*), but both clones expressed osteogenic markers (*Figure 1F*). Clone 3 lacked the chondrogenic markers, clone 4 displayed up-regulation of Sox9 (*Figure 1G*). Clone 3 expressed *Adiponectin,* while clone 4 did not (*Figure 1H*).

Based on ISCT criteria, for a clone to be considered as an MSC, it needed to demonstrate multipotent differentiation capacity in osteoblasts, chondrocytes and adipocytes (*Dominici et al., 2006*). Since we observed some level of disagreement between histological and molecular analysis of differentiation, we decided that each clone had to demonstrate positive histological staining with positive marker gene expression to be considered conclusively differentiated into a specific lineage. Therefore, for the eight clones presented in *Figure 1*, only normal clones 1, 2, and OA clones 1, 2 were defined as MSCs. *Supplementary files 4 and 5* summarize all data collected from both normal and OA knee joints, respectively. When the differentiation capacity of the clones was compared to the cells in situ (pre-culture) cell surface marker expression, the most frequent marker phenotype of cells that met the minimal criteria for MSCs (in situ) was $CD90^+CD44^+CD73^+$.

## Cell surface profile validation

To determine if $CD90^+CD44^+CD73^+$ expression defines MSC populations in situ, synovial membrane samples from a new cohort of patients (n=4 normal, n=4 OA) were obtained. In this experiment, we directly compared between cells that were triple positive for $CD90^+CD44^+CD73^+$ (live, $CD68^-$) vs. all live $CD68^-$ cells that did not have this exact expression profile. Interestingly, cells enriched for $CD90^+CD44^+CD73^+$ expression did not guarantee multipotent differentiation potential (summarized in *Supplementary file 6*). On the contrary, samples that did not contain the $CD90^+CD44^+CD73^+$ population still retained differentiation potential in OA and normal synovium meeting the MSC criteria. Furthermore, after culture in vitro, all lines expressed all MSC markers except CD271. Based on this experiment, an expression profile of $CD90^+CD44^+CD73^+$ on cells directly isolated from synovial membrane does not presumptively identify MSCs.

## In vivo functional assessment

It was next decided to investigate if there was a functional difference between MSC vs. non-MSC clonally derived populations when injected into the knees of rats with surgically induced OA. DMM surgery was performed on the left knee of Lewis rats and 1 week post-DMM surgery, rats were either injected with clones classified as MSCs or non-MSCs (from both normal and OA synovium). Histological sections from control (sham-injured) and injured (treated with saline) rats were compared to rats that had been injured and injected with either MSC or non-MSC clones (*Figure 2*). Knees that received sham DMM appeared to be morphologically normal (*Figure 2A*), while knees that underwent DMM and received saline showed proteoglycan loss, areas of cartilage loss, and osteophyte formation (*Figure 2B*). DMM knees injected with MSCs showed some areas of reduced proteoglycan staining and surface fibrillation but did not present with areas of cartilage loss (*Figure 2C*). Injured knees that received non-MSCs showed proteoglycan loss, areas of synovial inflammation, surface fibrillation, and areas of cartilage loss (*Figure 2D*). Osteoarthritis Research Society International (OARSI) histological scoring and Krenn scoring were undertaken to quantify OA severity and synovial inflammation, respectively. These scores demonstrated that rats receiving MSCs had a significantly lower OARSI and Krenn score vs. rats injected with non-MSCs post-DMM surgery (*Figure 2E*, *Figure 2—figure supplement 1*). These findings suggested that intra-articular injection of MSCs either inhibited degeneration and/or promoted regeneration to reduce the severity of OA, while the non-MSCs lacked this functional ability.

To examine whether the transplanted cell populations were capable of directly contributing to articular cartilage regeneration in vivo, the clones were virally labeled with TdTomato (to track their localization within the rat tissue). TdTomato$^+$ MSC clones (from normal and OA patients) were observed

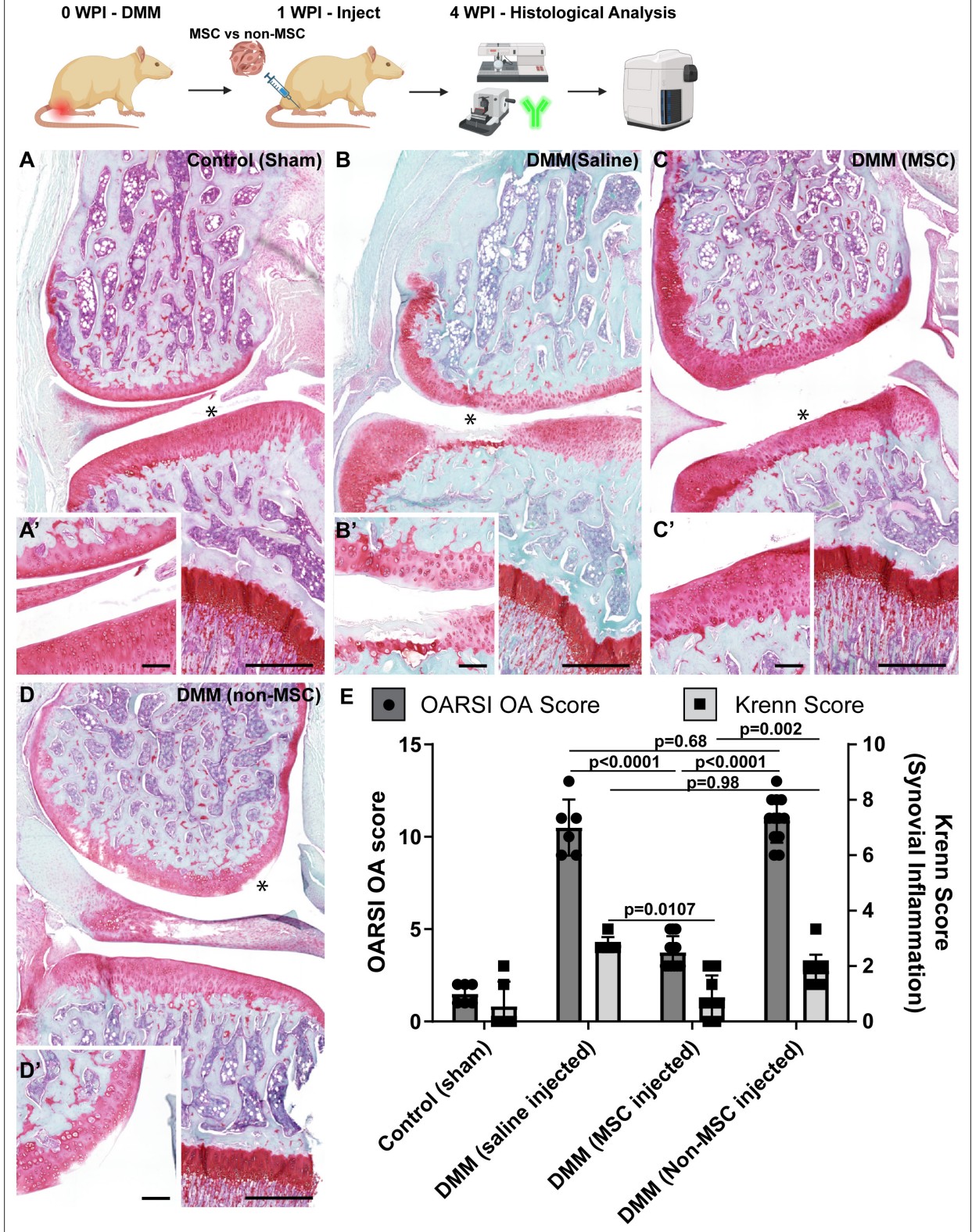

**Figure 2.** Osteoarthritis (OA) scoring with and without cell injection in a rat DMM model. Representative images of rats with sham surgery (**A, A'**), DMM injury with injection of saline (**B, B'**), DMM injury with injection of mesenchymal stem cells (MSCs) (**C, C'**) or non-MSCs (**D, D'**). The Osteoarthritis Research Society International (OARSI) and Krenn scoring was quantified in each group (**E**). Scale bars equal 200 μm in A, B, C, D and 20 μm in A', B', C', D'. *p<0.05. Sample size as follows: control (sham) n=6, DMM (saline injected) n=6, DMM (MSC injected) n=3 rats per cell line, eight cell lines used

*Figure 2 continued on next page*

*Figure 2 continued*

(normal MSC x2 and non-MSC x2, OA MSC x2 and non-MSC x2). Normal and OA MSC groups and non-MSC groups added together. Signifiance determined by 2 way ANOVA. *p*<0.05.

The online version of this article includes the following figure supplement(s) for figure 2:

**Figure supplement 1.** Safranin O staining sections of rat synovium with or without cell delivery.

within the articular cartilage (*Figure 3G, H, M and N*) while TdTomato[+] non-MSC clones (from normal and OA patients) were not observed with cartilage tissue (*Figure 3J, K, P and Q*). In uninjured rats (*Figure 3A and B*) or injured rats injected with saline alone (*Figure 3A and B*), no TdTomato signal was detected.

To determine if the transplanted human cells generated functional articular chondrocytes, Lubricin/PRG4 staining was undertaken, since this protein is produced by chondrocytes and is essential to joint homeostasis in part by lubricating the articular surfaces (*Das et al., 2019*; *Iqbal et al., 2016*). In control (sham) joints, lubricin staining is primarily confined to the superficial zone of the cartilage resulting in a continuous layer of expression (*Figure 3A and C*). In DMM joints injected with saline, this protective layer of PRG4 staining is disrupted (*Figure 3D and F*). When MSCs (from normal or OA synovium) were injected into DMM joints, some level of restoration of the PRG4 protective layer was observed (*Figure 3G, I, M and O*). Interestingly, it was observed that normal MSC clones throughout the cartilage produced PRG4 (*Figure 3G and I*), while this was not observed with OA MSC clones (*Figure 3M and O*). In DMM joints injected with non-MSC clones (normal or OA), the PRG4 protective layer was disrupted (*Figure 3J, K, P and R*), similar to saline-injected joints (*Figure 3D and F*). A tissue cytometry approach (*Figure 3—figure supplement 1*) was used to quantify the number of tdTomato, PRG4, and tdTomato with PRG4 expressing cells in the articular cartilage (*Figure 3S*). Overall, there was a significant increase in tdTomato and PRG4-expressing cells within the cartilage post-MSC treatment (regardless of being derived from normal/OA synovium); yet more tdTomato and PRG4-expressing cells were observed in joints treated with MSCs derived from normal vs. OA synovium. Furthermore, we also observed that while transplanted normal MSCs took on a PRG4-positive phenotype, MSCs from OA synovium did not. Non-MSCs, regardless of patient population, did not engraft into the articular cartilage nor take on a PRG4 positive phenotype (*Figure 3S*).

To determine if the MSC clones were regenerating/maintaining the articular cartilage in DMM joints, collagen 2 staining was undertaken. Collagen 2 staining was found within the articular cartilage of DMM joints and colocalized with TdTomato expression (*Figure 3—figure supplement 2*). Taken together, these outcomes suggest that intra-articular injection of MSCs (normal or OA) resulted in the regeneration of articular cartilage in comparison with the intra-articular injection of non-MSCs that had no beneficial effect on the articular cartilage.

## Localization of non-MSC clones within the joint

Since non-MSC clones were not observed within the articular cartilage of the rats, the synovial tissue was examined (*Figure 4*). Few MSCs (normal and OA – normal clone shown as a representative example) were found within the synovium or adjacent tissue (*Figure 4A*), while many non-MSCs (normal and OA – normal clone – same patient as MSC clone, shown as a representative example) were observed throughout the synovium in both the intimal and subintimal layers (*Figure 4B*). To determine what role the cells were playing in the micro-environment, CCL2 staining was performed. CCL2 is a pro-inflammatory chemokine which recruits circulating monocytes from the blood to infiltrate tissue during inflammatory processes *Deshmane et al., 2009*; is involved in OA pain and pathogenesis (*Jablonski et al., 2019*; *Miller et al., 2012*; *Raghu et al., 2017*). We have demonstrated that CCL2 is expressed in OA MSCs and regulates their behaviour (*Harris et al., 2013*). Although CCL2 staining did co-localize with tdTomato (MSCs), it was minimal (*Figure 4A*). CCL2 staining was also observed in the intimal layer but was independent of tdTomato expression (*Figure 4A*). This was in contrast to CCL2 staining when non-MSCs were injected into joints (*Figure 4B*). In non-MSC-injected joints, CCL2 staining was observed throughout the synovium and was highly co-localized with TdTomato expression (*Figure 4B*). Furthermore, since CCL2 is a secreted protein, it was interesting to observe that CCL2 staining appeared to diffuse out from the tdTomato[+] non-MSC clones (*Figure 4B*). When the number of CCL2-positive cells and the level of CCL2 staining (mean fluorescent intensity – MFI)

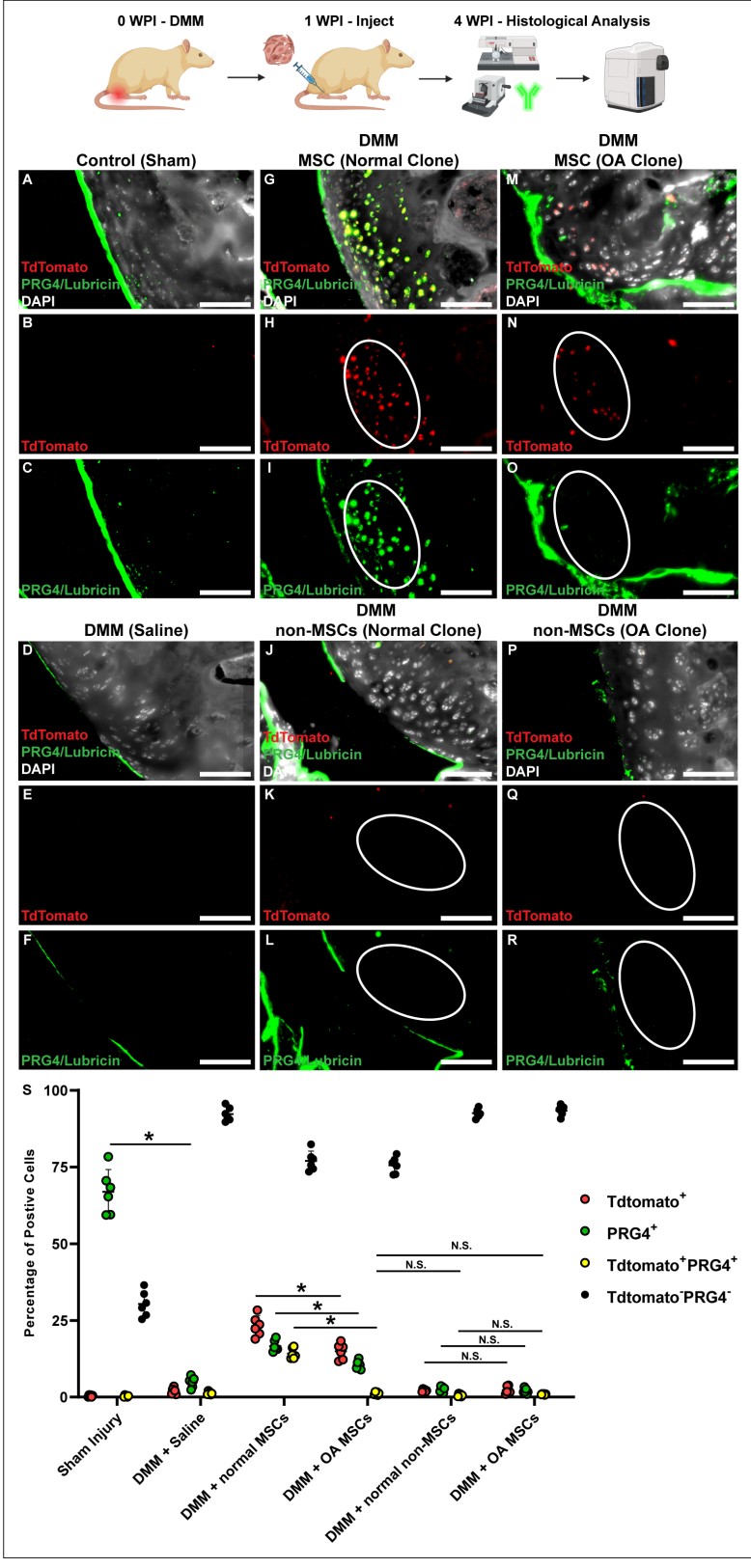

**Figure 3.** Exogenous cell contribution to cartilage repair in a rat DMM model. Representative images of rats with sham surgery (**A**) without cell injection (no TdTomato – B) and endogenous PRG4/Lubricin staining (**C**). In rats injected with saline (**D**), no TdTomato (**E**) and disrupted PRG4/Lubricin (**F**) staining was observed. When normal (**G**) or osteoarthritis (OA) (**M**), mesenchymal stem cells (MSCs) were injected into injured rats, TdTomato expression

*Figure 3 continued on next page*

*Figure 3 continued*

(H.N) was observed. When normal (**J**) or OA (**P**) non-MSCs were injected, no TdTomato (**K, Q**) and disrupted PRG4/ Lubricin (**L, R**) staining was observed. Scale bars equal 30 µm. Tissue cytometry was employed to quantify the number of tdTomato-positive, PRG4-positive, and double-positive cells within the articular cartilage (**S**). Signifiance determined by 2 way ANOVA. $*p<0.05$, $**p<0.01$. $*=p<0.05$, N.S.=not significant . Sample size as follows: control (sham) n=6, DMM (saline injected) n=6, DMM (MSC injected) n=3 rats per cell line, eight cell lines used (normal MSC x2 and non-MSC x2, OA MSC x2 and non-MSC x2).

The online version of this article includes the following figure supplement(s) for figure 3:

**Figure supplement 1.** Representative tissue cytometry gates.

**Figure supplement 2.** Mesenchymal stem cells (MSCs) within the cartilage produce collagen 2.

was quantified, it was found that significantly more CCL2-positive cells and expression was detected in the synovium of rats injected with non-MSCs (normal or OA) (*Figure 4C*).

In a previous study, we demonstrated that CCL2 acted upon MSCs by increasing their proliferative ability (*Harris et al., 2013*). Therefore, we decided to examine MSC/non-MSC proliferation in the joint (synovium and cartilage) through Ki67 staining (*Figure 4D–F*). Within the articular cartilage, transplanted MSCs (normal or OA) demonstrated little to no Ki67 staining, indicating minimal cell proliferation (*Figure 4D*). Since non-MSC clones were not observed in the articular cartilage, no Ki67 staining was observed. In the synovium, MSC clones (normal or OA) demonstrated minimal positive Ki67 staining, while non-MSC clones demonstrated robust staining (*Figure 4D*). However, it is important to remember that significantly fewer MSCs were observed within the synovial tissue (*Figure 4G*), yet these cells were proliferative.

## Proteomics analysis of clonal populations

Since a clear difference was observed between in vitro and in vivo functionality/behaviour in the two different populations of synovial cells, a quantitative shotgun proteomics analysis was undertaken to understand the proteomic differences between the subtypes (*Figure 5A*). Several differentially expressed proteins were identified between the subtypes (*Supplementary file 7*, *Figure 5B*). Using Metascape (*Figure 5C*) and STRING-db (*Figure 5D*), analyses were undertaken on the proteins differentially regulated in MSCs, and several pathways were differentially enriched. An aspect of this proteomic workflow that was interesting was that the MSC subtype was enriched for CD47, ITGA5, and DPP4 expression. While ITGA5 and DPP4 have already been implicated in fibroblast and progenitor function, CD47 has been traditionally thought of as a signal to immune/macrophages that blocks their phagocytosis of these CD47-expressing cells. In terms of MSC expression of CD47, this molecule has been shown to enhance homing and immunomodulatory ability of the cells. Therefore, we characterized freshly isolated synovial samples from normal (n=3) and OA (n=3) joints for CD90$^+$CD73$^+$CD44$^+$ cells and examined what percentage of this population expressed CD47.

## CD47 as a marker of human synovial cells with in vivo chondrogenic potential

Synovial biopsies were recovered from an additional n=3 normal and n=3 OA knee joints and the CD90$^+$CD73$^+$CD44$^+$ population was identified (*Figure 6A*). This population of cells was further interrogated for the expression of CD47, and it was found that all CD73$^+$CD44$^+$ double-positive cells expressed CD47, but there was also a CD47$^{Hi}$ population (*Figure 6B and C*) in both normal and OA synovium. The CD47$^{Hi}$ and CD47$^{Lo}$ sub-populations were both isolated by FACS and we further examined the immunophenotype of the cells. Specifically, we assayed if these CD47$^{Hi}$ and CD47$^{Lo}$ sub-populations expressed ITGA5 and/or DPP4/CD26 as these markers have previously been associated with stromal progenitors and shown to play a role in the progression of arthritis (*Zheng et al., 2025*). The CD47$^{Hi}$ sub-population from normal and OA synovium expressed both ITGA5 and DPP4/CD26, whereas the CD47$^{Lo}$ sub-population was largely negative for both markers (*Figure 6—figure supplement 1*). To further understand if there were any functional differences between sub-populations, we differentiated the cells into chondrocytes using standard pellet culture. Both CD47$^{Hi}$ and CD47$^{Lo}$ sub-populations produced pellets that stained positive for Alcian blue (*Figure 6B and D*); however, the CD47$^{Hi}$ cells appeared to have more instances of staining. Therefore, we quantified the GAG

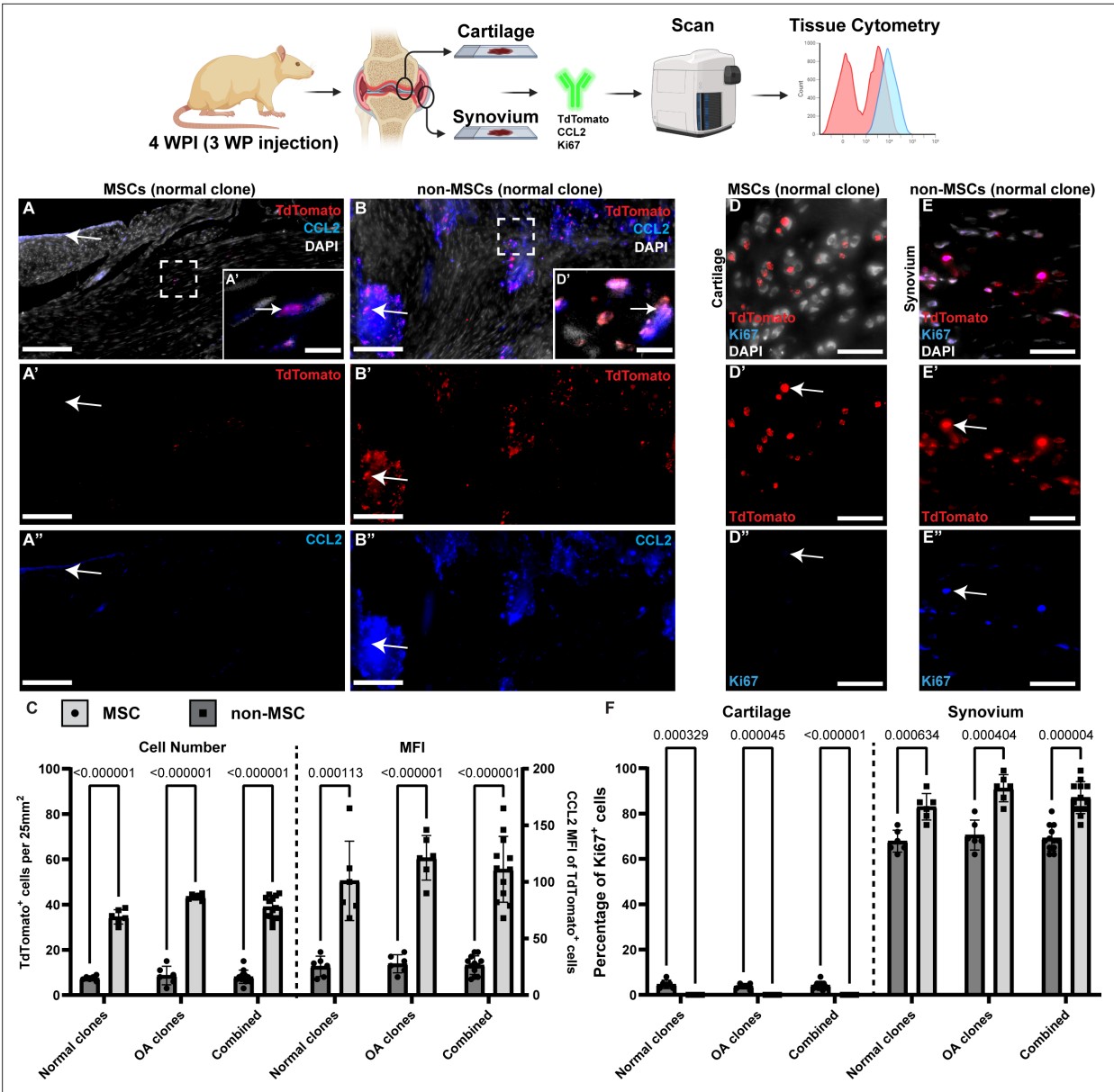

**Figure 4.** Contribution of transplanted cells to inflammatory signaling. Representative images of mesenchymal stem cells (MSCs) (**A**) and non-MSCs (**B**) in the synovium of rats post-DMM. In rats injected with MSCs, only minimal CCL2 (**A"**) expression is observed and/or colocalized with tdTomato staining (**A'**). In rats injected with non-MSCs, intense CCL2 staining (**B"**) was observed that colocalized with tdTomato staining (**B'**). This was quantified using tissue cytometry (**C**). In rats injected with MSCs, little Ki67 staining was observed in the cartilage (**D-D"**). In rats injected with non-MSCs, nearly every transplanted cell in the synovium was also positive for Ki67 (**E-E"**). This was quantified using tissue cytometry (**F**). Signifiance determined by t-Test. $p<0.05$. Scale bars equal 75 µm (**A-B**), 30 µm (**D-E**).

content and found that CD47$^{Hi}$ cells produced more GAGs than CD47$^{Lo}$ cells, regardless of if they were isolated from normal or OA synovium (**Figure 6E**). Collagen 2 staining of the pellets showed a similar trend in where more pronounced Col2 staining was observed in pellets derived from CD47$^{Hi}$ cells (**Figure 6F and G**).

These CD47$^{Hi}$ and CD47$^{Lo}$ populations were then injected into rats that had undergone DMM injury one week prior (**Figure 7A**). Rats that received CD47$^{Hi}$ cells presented with significantly lower OARSI scores than rats receiving CD47$^{Lo}$ cells (**Figure 7B–D**), but there was no difference in Krenn scoring between rats which received CD47$^{Hi}$ or CD47$^{Lo}$ cells (**Figure 7B**). When the joint tissues were isolated for histology, it was found that CD47$^{Hi}$ cells contributed to the formation of new articular cartilage tissue within the injured joints (**Figure 7H–J**); however, this was not observed with CD47$^{Lo}$ cells

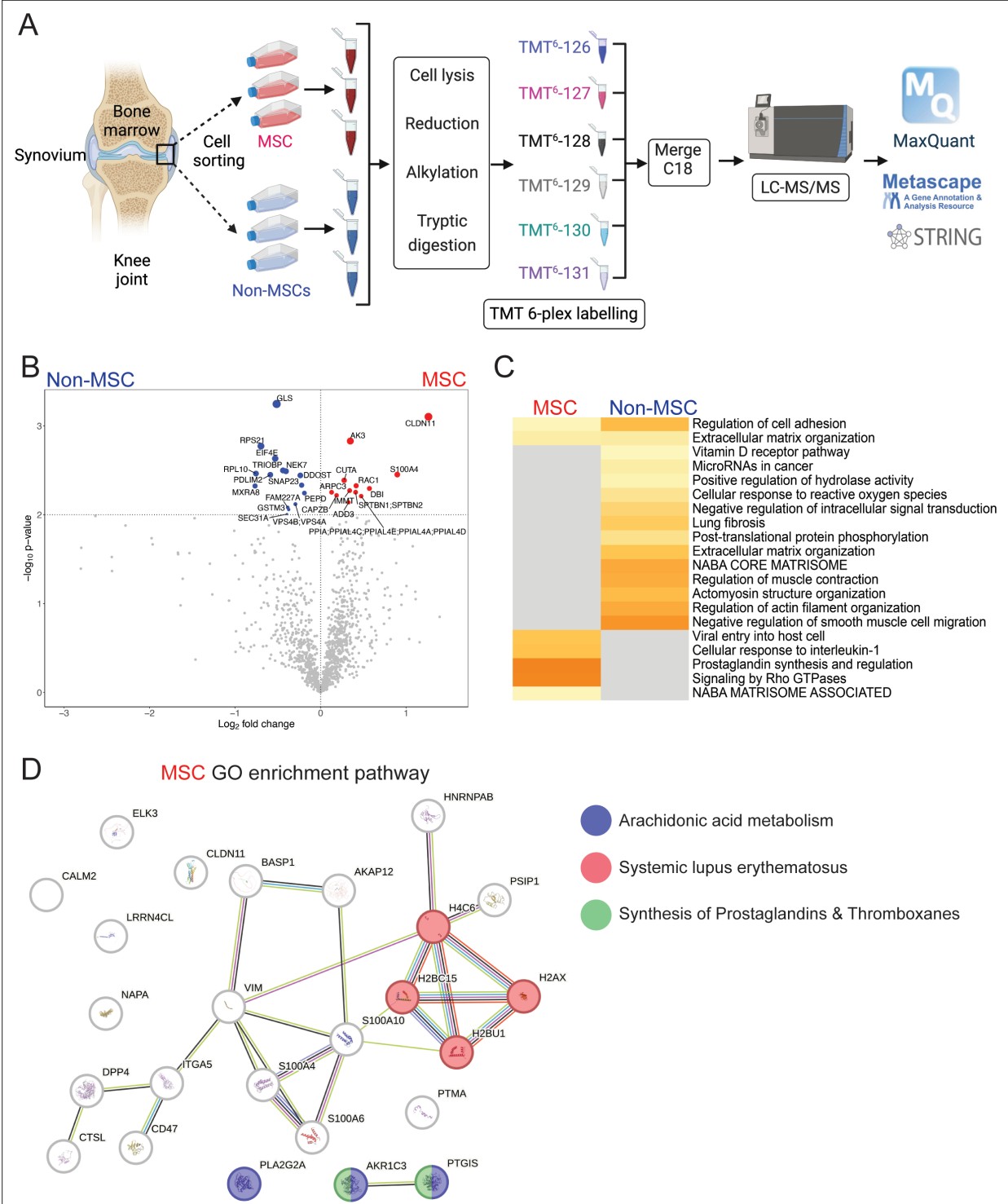

**Figure 5.** Proteomics analysis of mesenchymal stem cell (MSC) vs. non-MSC populations. Schematic representation of the quantitative shotgun proteomics workflow (**A**). Volcano plot demonstrating the distribution of statistically changing proteins identified in MSCs (red) and Non-MSCs (blue) (**C**). Metascape analysis and heatmap showing pathways regulated by differentially expressed proteins in MSCs vs. non-MSCs (**D**). STRING-db analysis of protein-protein interaction networks with elevated proteins in MSCs (**D**).

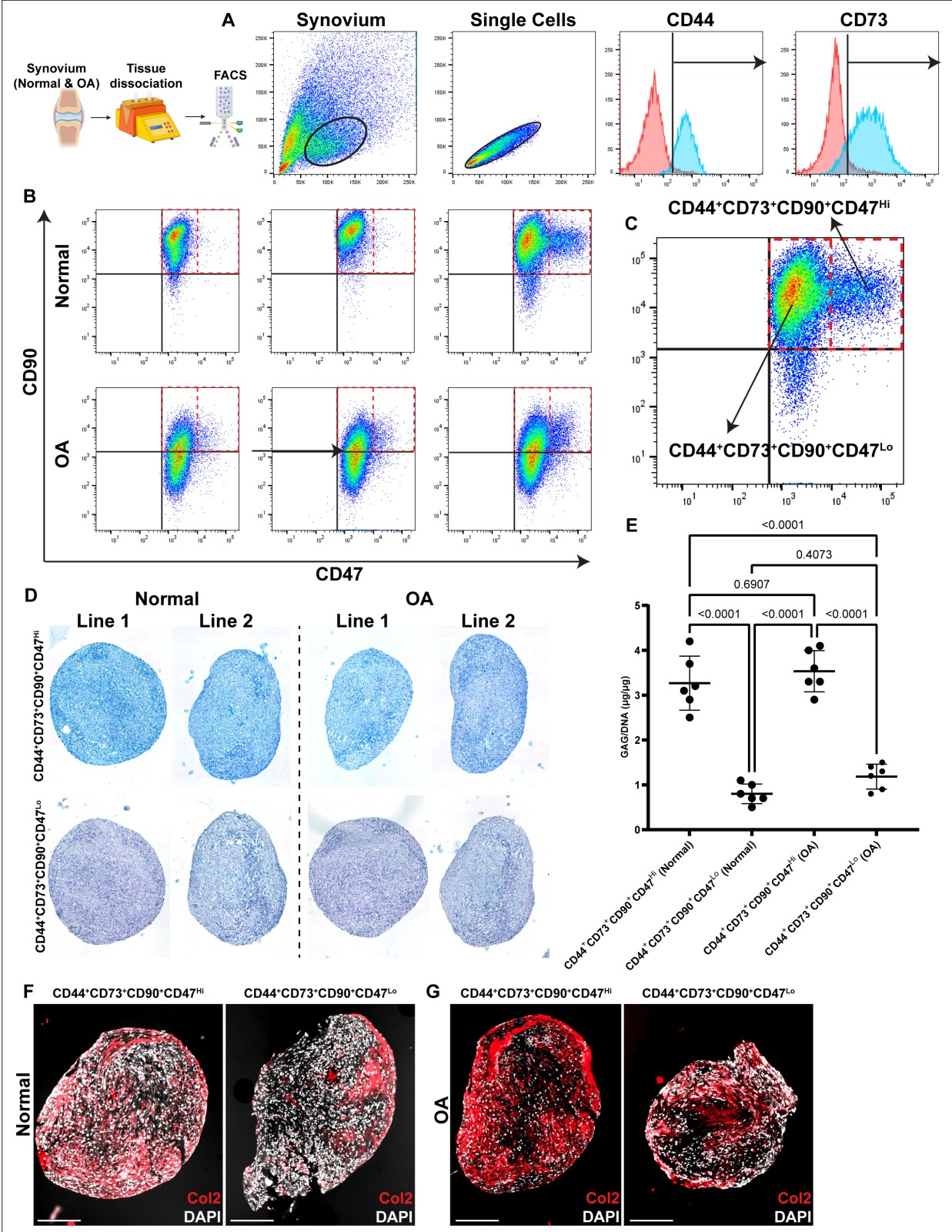

**Figure 6.** Analysis of synovial cells expressing CD47. The gating strategy to identify and sort CD47$^{Hi}$ vs. CD47$^{Lo}$ cell populations (**A–C**). CD47$^{Hi}$ vs. CD47$^{Lo}$ cells were identified and isolated from normal and osteoarthritis (OA) synovial tissue (**B**). Chondrogenic differentiation of CD47$^{Hi}$ vs. CD47$^{Lo}$ cells using pellet culture and stained with Alcian Blue (**D**). GAG quantification of the pellets (**E**). The pellets were also stained with Collagen 2 (Col2) as a marker of mature cartilage ECM (**E, G**). Signifiance determined by1 way ANOVA. $p<0.05$. Scale bars equal 100 µm.

*Figure 6 continued on next page*

*Figure 6 continued*

The online version of this article includes the following figure supplement(s) for figure 6:

**Figure supplement 1.** Analysis of synovial cells expressing CD47.

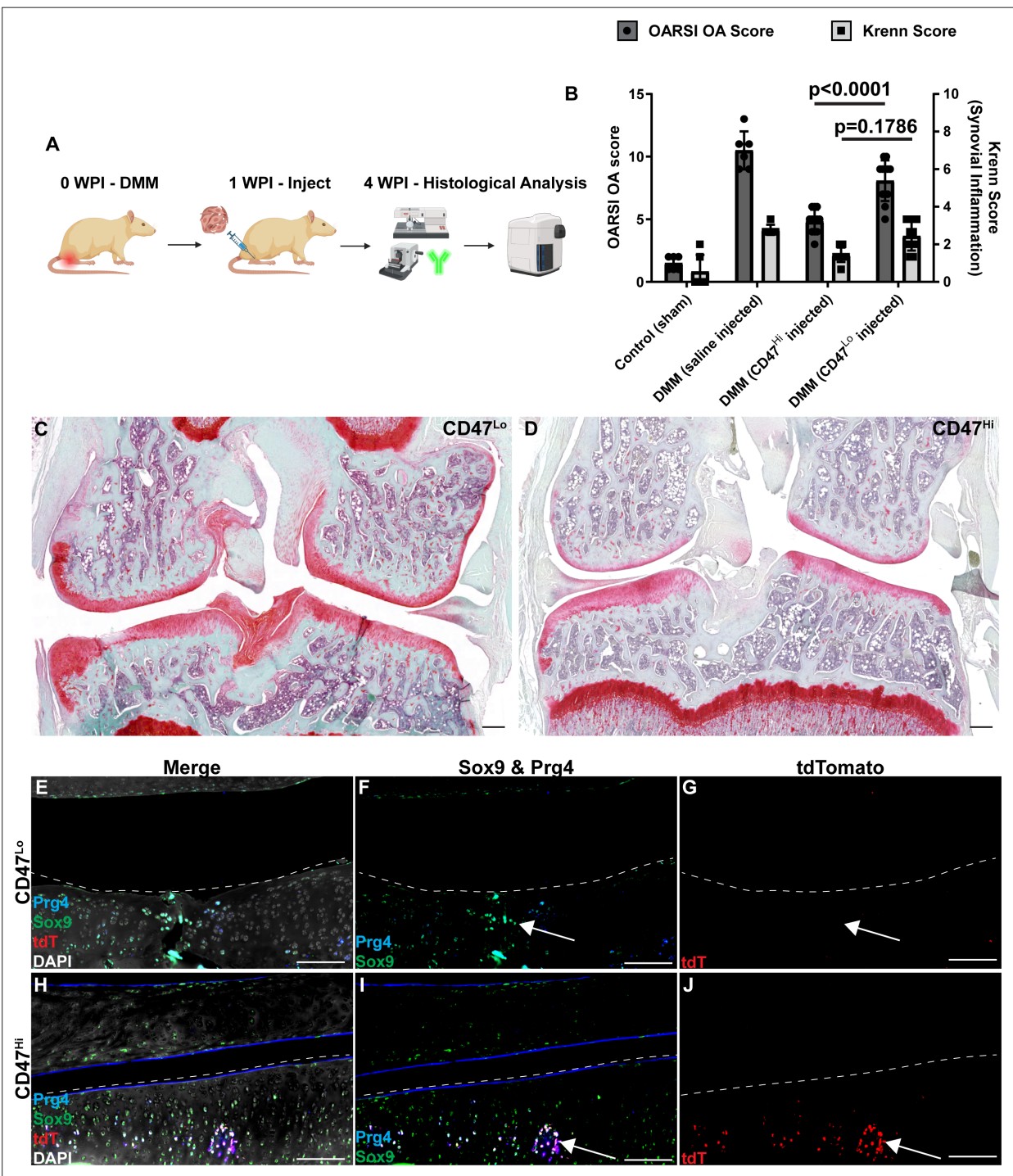

**Figure 7.** CD47$^{Hi}$ vs. CD47$^{Lo}$ cell contribution to cartilage repair in a rat DMM model. The CD47$^{Hi}$ vs. CD47$^{Lo}$ cells were injected into rats that underwent DMM injury (**A**), and it was observed that rats which received the CD47$^{Hi}$ cells presented with significantly lower OARSI scores than rats receiving CD47$^{Lo}$ cells (**B**) with representative images presented of joints receiving CD47$^{Lo}$ (**C**) and CD47$^{Hi}$ (**D**) cells. Immunofluorescence analysis demonstrated that the transplanted human CD47$^{Hi}$ cells (TdTomato) integrated into the rat articular cartilage and expressed the chondrocyte markers Prg4 and Sox9 (**H–J**). This was not observed in the cartilage of rats that received the CD47$^{Lo}$ cells (**E–G**). Scale bars equal 50 μm. Signifiance determined by1 way ANOVA. *p*<0.05.

(*Figure 7E–G*). Specifically, human cells (tdTomato[+]) were observed within the cartilage of CD47[Hi]-injected rats, and these cells also expressed the chondrogenic markers Sox9 and Prg4 (*Figure 7H–J*). In rats that received CD47[Lo] cells, few to no tdTomato cells were observed within the injured cartilage (*Figure 7E–G*).

## Discussion

The synovium plays a vital role in the overall health of the joint (*Berenbaum, 2013*; *Kapoor et al., 2011*). In a normal, healthy joint, synoviocytes assist in nourishing chondrocytes, removing waste and debris, and facilitating gas exchange. However, in OA, the breakdown of the cartilaginous matrix and subsequent phagocytosis of debris by synoviocytes promotes synovitis. This inflammation triggers the release of soluble proinflammatory mediators from the synovium, enhancing cartilage breakdown and perpetuating a vicious cycle. Although OA disrupts the primary function of articular cartilage (to provide cushioning and decrease friction within the joint), it impacts all tissues within the joint environment. The loss of the protective cartilage layer leads to intense pain in affected individuals. Moreover, damaged articular cartilage has limited ability to repair itself due to its avascular and aneural nature (*Masson and Krawetz, 2020*). Current non-surgical treatment strategies for OA focus on alleviating symptoms, such as pain and inflammation. However, there are currently no proven treatments to prevent cartilage loss and promote tissue regrowth. While MSCs from the synovial membrane and synovial fluid have shown tri-lineage differentiation potential in various studies, endogenous articular cartilage repair remains elusive (*De Bari, 2015*; *Sekiya et al., 2021*; *Krawetz et al., 2012*; *Masson et al., 2015*; *Sekiya et al., 2012*). Moreover, distinct sub-populations of synovial MSCs exhibit widely different characteristics, including cell surface protein expression and differentiation potential (*Affan et al., 2019*). This suggests that not all tissue-resident stem cells are created equal and underscores the need for further research into understanding these sub-populations.

The aim of this project was to characterize different progenitor populations in the synovium of normal and OA knee joints based on their self-renewal capacity, differentiation ability, and cell surface marker profile. However, relying solely on cell surface markers for MSC selection can be unreliable, as not all subpopulations demonstrate a high correlation between marker expression and differentiation potential. Moreover, changes in marker expression under different laboratory conditions can confound MSC characterization. Furthermore, it has been shown that some cell surface markers (such as CD105) may be an artifact of in vitro culturing, as this molecule doesn't appear to be expressed by MSCs in vivo and is in part responsible for the ability of MSCs to adhere to plastic (*Anderson et al., 2013*; *Cleary et al., 2016*). Therefore, our study utilized single-cell sorting to isolate cells from human knee samples, minimizing exposure to the culture microenvironment before analysis. We found that regardless of in situ marker expression, nearly all clonal cell populations displayed a uniform marker profile consistent with a presumptive MSC phenotype. This strongly suggests that the in vitro microenvironment upregulates and/or stabilizes the expression of the receptors. This notion is supported by our observation (similar to previously published studies) (*Affan et al., 2019*), that the expression of CD105 was acquired only after culture, indicating its potential role as a bystander marker in MSC characterization. Additionally, the expression of CD271 varied among MSC clones, with none of the clones demonstrating multipotential differentiation capacity expressing CD271. This suggests that CD271 may not be a marker of synovial MSCs (*Barilani et al., 2018*). Also interesting was that the distribution of OA MSC subpopulations differed from those in healthy joints, indicating distinct alterations in the synovial niche during OA onset/progression. In future studies, it would be interesting to determine what exact aspects of in vitro culture cause the cells to take on a uniform immunophenotype. Specifically, testing out different media and/or matrix compositions could shed light on this observation.

In the current study, we have identified a subpopulation of synovial cells that express high levels of CD47. Our findings support the hypothesis that cells expressing high levels of CD47 have robust cartilage regeneration potential. CD47, also known as integrin-associated protein, is a transmembrane protein that plays a pivotal role in immune regulation. It acts as a 'don't eat me' signal, inhibiting phagocytosis by macrophages and other immune cells (*Hao et al., 2023*; *Maute et al., 2022*; *Oldenborg, 2013*). It has been shown that overexpression of CD47 allows cells (such as in various cancers) to evade immune surveillance. In stem cell populations, particularly adult stem cells, CD47 plays a crucial role in maintaining self-renewal and tissue regeneration capacities (*Jaiswal et al., 2009*; *Kim et al., 2021*). It regulates stem cell survival and proliferation by interacting with its receptor, SIRPα,

on neighboring cells or stromal components. This interaction influences stem cell behavior, homing, and differentiation, shaping tissue homeostasis and repair processes. Its upregulation on dying cells can lead to the inhibition of phagocytosis of these necrotic/apoptotic cells (*Kojima et al., 2016*), leading to the accumulation of cellular debris within the environment, thus perpetuating inflammation. Overall, there is still not much known about how CD47 regulates stem cell/progenitor function, and it would be of interest to experimentally test if the overexpression of CD47 in MSCs can increase their endogenous cartilage regeneration capacity.

While the clonal derivation and long-term expansion of cells made this experimental design possible, this approach is not without limitations. Firstly, during clonal derivation, cells may acquire genetic mutations or epigenetic alterations due to the stresses of isolation and culture conditions, leading to phenotypic changes and potential change/loss of functionality. Moreover, the process of long-term expansion in vitro can induce replicative senescence, telomere shortening, and genomic instability, compromising the quality and integrity of the cell population over time. This could account in part for the majority of clones we derived losing the ability to proliferate before we could achieve a sufficient number of cells for the downstream analyses. Furthermore, cells cultured in vitro often face challenges in recapitulating the complex microenvironment and dynamic interactions present with their in vivo niche(s), potentially altering their behavior and functional properties. As such, the reliance on exogenous growth factors, culture media, and artificial substrates to support cell growth and expansion introduces variability and may not fully mimic the physiological cues present in the native tissue environment. Such discrepancies can impact the reliability and reproducibility of experimental outcomes and may hinder the successful clinical translation of cell-based therapies. Yet, even with these challenges, we have presented compelling evidence demonstrating that a sub-population of cells exists within the human synovial membrane (under normal and OA conditions) that can contribute to articular cartilage regeneration.

In conclusion, while MSCs hold promise for OA treatment, the heterogeneity within MSC populations presents challenges for their effective utilization. The development of effective and reproducible protocols for MSC identification/characterization are necessary to ensure the selection of MSCs with the greatest potential for joint repair and regeneration. With this knowledge, we have the best chance of developing novel cell therapies that will benefit the growing number of patients suffering from OA.

## Methods

### Study participants

This study protocol was approved by the University of Calgary Human Research Ethics Board (REB15-0005 and REB15-0880). The n=18 (with an additional n=4 for the validation phase) normal control cadaveric donations were harvested within 4 hr of death and were at least 18 years old with no history of arthritis, joint injury or surgery, no prescription anti-inflammatory medications, and no co-morbidities. The n=15 (with an additional n=4 for the validation phase) knee OA sample donors were diagnosed by an orthopedic surgeon at the University of Calgary based on clinical symptoms with radiographic evidence in accordance with American College of Rheumatology criteria (*Supplementary file 1*). All OA participants provided written consent to participate. All testing was carried out in accordance with the Declaration of Helsinki.

### Experimental outline

The experimental outline of the study is presented in *Figure 1—figure supplement 3*.

### Cell sorting and flow cytometry

Synovial cells were isolated as previously described (*Affan et al., 2019*). Briefly, 5 mm² tissue samples were digested using 1 mg/mL Collagenase IV (Thermo Scientific) at 37 °C with shaking for 90 min. The resulting cell suspension was filtered using a 70 µm filter and centrifuged at 5000 rpm for 6 min. The cell pellet was then washed with 1 x PBS (Lonza-BioWhittaker) and re-suspended in 100 µL of MesenCult Media (StemCell Technologies).

Cells were immunostained for mesenchymal stem cell (MSC) markers, including CD90 (Clone 5E10, PE), CD271 (Clone C40-1457, BV421), CD105 (Clone 266, BV650), CD73 (Clone AD2, APC), and CD44 (Clone G44-26, PE-Cy7) (all from BD Bioscience), according to the manufacturers' protocols.

Additionally, cells were stained with the viability marker FVS510 (BV510) at a 1:100 dilution for 30 min on ice and with CD68 (Clone Y1/82 A, FITC) to target the macrophage population. Positive compensation controls were conducted using Ultracomp eBeads (eBioscience), and unstained cells served as negative controls.

The labeled cells were sorted using a BD FACS Aria Fusion with an indexed sorting protocol. The instrument was configured with a 100 µm sorting nozzle, a 'single cell' mask, a 2 X neutral density filter, and a flow rate below 50% of the maximum. Dead cells (FVS510+) and macrophages (CD68+) were excluded, and the remaining cells were collected into 96-well plates (1 cell per well) containing Dulbecco's modified Eagle's medium F-12 (DMEM/F-12) with MesenCult-SF attachment substrate (StemCell Technologies) and 1% antibiotic-antimycotic (Life Technologies). The sorted cells were cultured until they reached 70–80% confluency, then transferred to T-25 flasks (Greiner). Once the cultures reached ~70% confluency, the cells were washed with 1 x PBS (Lonza-BioWhittaker), trypsinized (Corning), and re-plated. Media was changed every 2 days, and cells were passaged until they expanded into 4 x T-75 flasks.

During the validation phase of the study, bulk sorting was performed directly from dissociated synovium without prior cell expansion. Cell suspensions from a new cohort (n=4 normal and n=4 osteoarthritis synovium) were isolated using the same digestion method described earlier. The cells were stained with the markers CD68, CD90, CD73, CD44, and the viability marker FVS510, as previously detailed. Sorting was performed using the BD FACS Aria Fusion, and cells were sorted into 5 mL tubes containing 500 µL of media. Instead of sorting one cell per well, the strategy involved gating out CD68 + and dead cells, followed by sorting CD90+CD73+CD44+ cells into one tube and the remaining cells (a mixture of all populations except CD90+CD73+CD44+) into a separate tube. This sorting was conducted using a 100 µm sorting nozzle, a 'purity' mask, a 2 X neutral density filter, and a flow rate of 5/11. Both positive and negative populations were transferred to 48-well plates and expanded until they reached 70% confluency in 4 x T-75 flasks. These cells were subsequently characterized for their differentiation potential and underwent in vitro flow cytometry analysis as previously described.

## Differentiation analysis

Expanded clonal cells were induced to differentiate into bone, cartilage, and fat as previously (*Affan et al., 2019*; *Krawetz et al., 2022*). For each differentiation protocol, $5 \times 10^5$ cells were either plated (for osteogenic and adipogenic protocols) or pelleted and plated (for chondrogenic protocols) in 24-well plates. Cells were incubated for 21 days in either osteogenic, chondrogenic, or adipogenic media, with the media changed every 2 days: **Osteogenic media**: DMEM/F-12, 10% FBS, 1% antibiotic-antimycotic, 1% MEM non-essential amino acids (NEAA), $10^{-4}$M dexamethasone, 50 µg/mL L-ascorbic acid (AA), and 10 mM β-glycerophosphate (all from Sigma). **Chondrogenic media**: DMEM/F-12, 10% FBS, 1% antibiotic-antimycotic, 1% MEM NEAA (Gibco), 10 nM dexamethasone, 50 mg/mL AA, NaOH, 10 ng/mL TGF-β3 (Peprotech), 500 ng/mL BMP-2 (Peprotech), sodium pyruvate (Gibco), and insulin-transferrin-selenium (ITS, Lonza-BioWhittaker). **Adipogenic media**: DMEM/F-12, 10% FBS, 1% antibiotic-antimycotic, 1% MEM NEAA, 1 µM dexamethasone, 10 µM insulin, 200 µM indomethacin, and 500 µM isobutylmethylxanthine (all from Sigma).

## Quantitative PCR (qPCR)

Cells were washed with 1 X PBS. RNA was extracted using TRIzol reagent (Invitrogen) from osteogenic and adipogenic cells, and using the Total RNA Kit I (OMEGA bio-tek) from chondrogenic cells, following manufacturers' protocols. The isolated mRNA was stored at –80 °C for later analysis. To generate cDNA, 10 µL of mRNA was mixed with 10 µL of cDNA Master Mix (High-Capacity cDNA kit, Applied Biosystems) and incubated in a thermocycler (Bio-Rad) for 2 hr. The samples were then stored at –20 °C. qPCR was used to quantify the gene expression of specific markers: **Osteoblast markers**: *Sp7*, *Runx2*, **Chondrocyte markers**: *Sox9*, *Col2a*, **Adipocyte markers**: *Adipoq*. Triplicate reactions per sample were performed in either 384-well or 96-well plates (Applied Biosystems). Reactions included 0.5 µL of the specific probe, 5 µL of TaqMan Universal PCR Master Mix (Applied Biosystems), 3.5 µL ultrapure $H_2O$, and 1 µL cDNA. Gene expression was standardized using the 18 S ribosomal RNA, calculated via the DeltaDeltaCT method.

## In vitro histology

Cells were fixed with 10% neutral buffered formalin (NBF) at room temperature for 1 hr. **Osteogenic cultures** were stained with 20 mg/mL alizarin red (Sigma) for 45 min. **Adipogenic cultures** were washed, incubated with 60% isopropanol, and stained with Oil Red O for 15 min, followed by Harris Hematoxylin. **Chondrogenic cultures** were treated with 1% acetic acid and stained with 0.1% Safranin O for 5 min as previously described (*Affan et al., 2019*).

## Destabilization of the medial meniscus (DMM) injury model

Animal studies were carried out in accordance with the recommendations in the Canadian Council on Animal Care Guidelines. Animal protocols and surgical procedures in this study were approved by the University of Calgary Health Sciences Animal Care Committee.

Ten-week-old Lewis rats were used for the DMM model. Males weighed ~250 g (±50 g), and females weighed ~175 g (±20 g) at the time of surgery. Rats were anesthetized with isoflurane, and a medial para-patellar arthrotomy was performed under a microscope. The fat pad over the cranial horn of the medial meniscus was retracted, and sectioning of the medial meniscotibial ligament destabilized the meniscus (*Das et al., 2023*; *Iqbal et al., 2016*). Sham-operated rats served as controls, receiving surgery without injury. The joint capsule and skin were closed with sutures and adhesive.

## Cell injection

Once a sufficient number of cells had been obtained, the purified synovial cells were incubated with the following mix overnight (~12 hr) at 37 °C, 2% $O_2$: 5 mL MesenCult, 5 µL tdTomato lentivirus (where tdTomato was driven by the EF1-alpha promoter), and 2 µL of Polybrene (8 µg/mL, Sigma). Medium was changed the following day. The cells were then sorted for tdTomato expression and the positive cells were used for the xenotransplantation experiments.

Control groups included six rats with sham surgery and six rats that underwent DMM injected with PBS alone. The injection involved a 30 G needle placed through the patella tendon into the joint space, with 10 µL of sterile DPBS containing 100,000 cells injected 1 week post-DMM surgery. Sample size as follows for the initial experiments: DMM (MSC injected) n=3 rats per cell line, 8 cell lines used (normal MSC x2 and non-MSC x2, OA MSC x2 and non-MSC x2). Sample size as follows for the CD47 experiments: DMM (CD47$^{Hi}$ injected) n=2 rats per cell line, six cell lines used (three normal, three OA), (CD47$^{Lo}$ injected) n=2 rats per cell line, six cell lines used (three normal, three OA).

## Histology and OA grading

Four weeks post-cell injection (5 weeks post-DMM), rats were sacrificed. Knee joints were dissected, fixed in 4% formalin, decalcified, embedded in paraffin, and sectioned. Sections were stained with Safranin O to visualize proteoglycans and graded according to OARSI guidelines for rat knees.

Immunohistochemistry analysis was performed on the rat knee sections. Antigen retrieval was achieved using 10 mM sodium citrate (pH 6.0), and non-specific blocking was prevented using goat serum (1:500 dilution in TBST). tdTomato (AB8181, Origene), Collagen 2 (Col2; Clone # II-II6B3, DSHB), PRG4 (Clone # 9G3, Millipore), Ccl2 (Clone # 2D8, Thermo Fisher), Ki67 (Clone # SolA15, ThermoFisher), Sox9 (Clone # 7H13L8, Thermo Fisher) or CD47 (Clone # B6H12, Thermo Fisher) were applied to the sections and incubated overnight. For the Col2 staining, an additional hyaluronidase (Sigma) treatment step was added. Secondary controls were also performed, where only secondary antibody was applied to the sections (no primary antibody). All slides were mounted using EverBrite Hardset Mounting Medium with 4',6-diamidino-2-phenylindole (DAPI, Biotium). Slides were imaged using a Plan-Apochromat objective on an Axio Scan.Z1 Slide Scanner microscope (Carl Zeiss).

## Tissue cytometry

TissueQuest software was used for immunofluorescent image analysis. After nuclear segmentation using DAPI, each marker channel was analyzed, with gates and thresholds set using non-stained controls. Data were visualized using scattergrams and histograms, and statistical analysis was done using GraphPad.

## Quantitative shotgun proteomics using Tandem Mass Tags (TMT-6) labeling

Synovial membrane-derived clones were lysed and sonicated. Cellular proteins were quantified and isolated for TMT-6 plex shotgun proteomics. LC-MS/MS analysis was conducted at the Southern Alberta Mass Spectrometry core facility (*Das et al., 2023*). Peptides were separated using C18 columns, and data were analyzed using MaxQuant software and statistical tools in R.

## Reactome pathway analysis

The STRING database was used to identify protein interconnectivity, with analysis done in Metascape for functional enrichment, interactome analysis, and gene annotation. Human protein data were analyzed with a 1% false discovery rate.

## Statistics

Statistical analysis was conducted using GraphPad Prism 7. Data were presented as mean ± standard deviation (SD). The Student's t-test was used for group comparisons, and Spearman correlation was applied for association analyses. Significance was set at $p < 0.05$, with data visualized using IBM SPSS.

## Acknowledgements

The authors would like to thank Dr. Yiping Liu of the Cumming School of Medicine Flow Cytometry Facility for assistance with FACS and the Cumming School of Medicine ARC staff for assistance with animal husbandry. Natural Sciences and Engineering Research Council (NSERC) of Canada RGPIN-2014–04586 (RJK). Canada Foundation for Innovation (RJK). Calgary Foundation, Grace Glaum Professorship (RJK). The funders had no role in study design, data collection and analysis, decision to publish, or preparation of the manuscript.

## Additional information

### Funding

| Funder | Grant reference number | Author |
|---|---|---|
| Natural Sciences and Engineering Research Council of Canada | RGPIN-2014-04586 | Roman J Krawetz |
| Canada Foundation for Innovation | JELF | Roman J Krawetz |
| Calgary Foundation | Grace Glaum Professorship | Roman J Krawetz |

The funders had no role in study design, data collection and interpretation, or the decision to submit the work for publication.

### Author contributions

Nedaa Al-Jezani, Conceptualization, Formal analysis, Investigation, Methodology, Writing – original draft; Asmaa Affan, Formal analysis, Investigation, Methodology, Writing – review and editing; Catherine Leonard, Formal analysis, Investigation, Methodology; Nabangshu Das, Luiz Gustavo Almeida, Daniel Young, Paul Salo, Methodology, Writing – review and editing; Anand O Masson, Conceptualization, Investigation, Methodology, Writing – review and editing; Antoine Dufour, Supervision, Methodology, Writing – review and editing; Pam Railton, James N Powell, Investigation, Methodology, Writing – review and editing; Roman J Krawetz, Conceptualization, Resources, Formal analysis, Supervision, Funding acquisition, Investigation, Writing – original draft, Project administration, Writing – review and editing

### Author ORCIDs

Luiz Gustavo Almeida https://orcid.org/0000-0001-8961-8883
Paul Salo https://orcid.org/0000-0002-4005-9758

Roman J Krawetz (iD) https://orcid.org/0000-0002-2576-4504

### Ethics

Human subjects: This study protocol was approved by the University of Calgary Human Research Ethics Board (REB15-0005 and REB15-0880). All OA participants provided written consent to participate. Animal studies were carried out in accordance with the recommendations in the Canadian Council on Animal Care Guidelines. Animal protocols and surgical procedures in this study were approved by the University of Calgary Health Sciences Animal Care Committee (Ethics ID AC20-0042).

Reviewer #1 (Public review): https://doi.org/10.7554/eLife.103332.3.sa1
Reviewer #2 (Public review): https://doi.org/10.7554/eLife.103332.3.sa2
Author response https://doi.org/10.7554/eLife.103332.3.sa3

---

## Additional files

### Supplementary files

Supplementary file 1. Summary of the participants in the study. There was no difference in the ages between the normal and OA group ($P$=0.13).

Supplementary file 2. Summary of all clonal lines derived from all patients included in the study. Gray highlight: Patient sample did not yield any cell growth following single cell sorting. Blue highlight: Clonal cell line yielded enough number of cells for full characterization.

Supplementary file 3. Summary of the self-renewal capacity (population doublings) from all clonal lines derived in the study.

Supplementary file 4. Summary of the cell surface marker expression (in situ and in vitro) and differentiation potential of clones derived from normal individuals.

Supplementary file 5. Summary of the cell surface marker expression (in situ and in vitro) and differentiation potential of clones derived from OA patients.

Supplementary file 6. Summary of marker expression and differentiation potential from $CD90^+CD44^+CD73^+$ cells derived from normal and OA synovium.

Supplementary file 7. Shotgun proteomics of proteins from mesenchymal stem cells and non-mesenchymal stem cells identified without constraint by enzyme specificity rules during spectrum-to-sequence matching.

MDAR checklist

### Data availability

Data available within the article or its supplementary materials.

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
