## [Editor Report · eLife Assessment]

This **important** study characterized and identified clonal MSC populations from human synovium. The authors provide **convincing** evidence that clonal MSC populations can be isolated and expanded from both normal and osteoarthritic synovium and that CD47 represents a potential marker for improved chondrogenic potential of MSC sub-populations. These findings could provide new avenues for osteoarthritis treatment in the future and deeper mechanistic understanding of the factors involved in the repair.

---

## [Referee Report · Reviewer #1 (Public review)]

Summary:

This work by Al-Jezani et al. focused on characterizing clonally derived MSC populations from the synovium of normal and osteoarthritis (OA) patients. This included characterizing the cell surface marker expression in situ (at time of isolation), as well as after in vitro expansion. The group also tried to correlate marker expression with trilineage differential potential. They also tested the ability of the different sub-populations for their efficacy in repairing cartilage in a rat model of OA. The main finding of the study is that CD47hi MSCs may have a greater capacity to repair cartilage than CD47lo MSCs, suggesting that CD47 may be a novel marker of human MSCs that have enhanced chondrogenic potential.

Strengths:

Studies on cell characterization of the different clonal populations isolated indicate that the MSC are heterogenous and traditional cell surface markers for MSCs do not accurately predict the differentiation potential of MSCs. While this has been previously established in the field of MSC therapy, the authors did attempt to characterize clones derived from single cells, as well as evaluate the marker profile at the time of isolation. While the outcome of heterogeneity is not surprising, the methods used to isolate and characterze the cells were well developed. The interesting finding of the study is the identification of CD47 as a potential MSC marker that could be related to chondrogenic potential. The authors suggest that MSCs with high CD47 repaired cartilage more effectively than MSC with low CD47 in a rat OA model.

Comments on revisions:

Thank you for addressing the comments from the first review. No additional revisions.

---

## [Referee Report · Reviewer #2 (Public review)]

Summary:

This is a compelling study that systematically characterized and identified clonal MSC populations derived from normal and osteoarthritis human synovium. There is immense growth in the focus on synovial-derived progenitors in the context of both disease mechanisms and potential treatment approaches, and the authors sought to understand the regenerative potential of synovial-derived MSCs.

Strengths:

This study has multiple strengths. MSC cultures were established from an impressive number of human subjects, and rigorous cell surface protein analyses were conducted, at both pre-culture and post-culture timepoints. In vivo experiments using a rat DMM model showed beneficial therapeutic effects of MSCs vs non-MSCs, with compelling data demonstrating that only "real" MSC clones incorporate into cartilage repair tissue and express Prg4. Proteomics analysis was performed to characterize non-MSC vs MSC cultures, and high CD47 expression was identified as a marker for MSC. Injection of CD47-Hi vs CD47-Low cells in the same rat DMM model also demonstrated beneficial effects, albeit only based on histology. A major strength of these studies is the direct translational opportunity for novel MSC-based therapeutic interventions, with high potential for a "personalized medicine" approach.

Weaknesses:

Weaknesses of this study include the rather cursory assessment of the OA phenotype in the rat model, confined entirely to histology (i.e. no microCT, no pain/behavioral assessments, no molecular readouts). This is relevant given the mixed results in therapeutic experiments demonstrating lower OA scores, but not lower inflammation scores, in CD47-Hi-treated rats. Thus, future work should focus on characterizing the therapeutic mechanism further given the clinical relevant of inflammation and pain in OA. It is somewhat unclear how the authors converged on CD47 vs other factors, but despite its somewhat broad profile, it was shown to be a useful marker to differentiate functional effects of MSCs. Additional work is needed to understand whether MSCs also engraft in ectopic cartilage (in the context of osteophyte/chondrophyte formation) or whether their effects are limited to articular cartilage. Despite these areas for improvement, this is a strong paper with a high degree of rigor, and the results are compelling, timely, and important.

Overall, the authors achieved their aims, and the results support not just the therapeutic value of clonally-isolated synovial MSCs but also the immense heterogeneity in stromal cell populations (containing true MSCs and non-MSCs) that must be investigated further. Of note, the authors employed the ISCT criteria to characterize MSCs, with mixed results in pre-culture and post-culture assessments. This work is likely to have a long-term impact on methodologies used to culture and study MSCs, in addition to advancing the field's knowledge about how synovial-derived progenitors contribute to cartilage repair in vivo.

Comments on revisions:

I commend the authors for a good revision. While the revision primarily entailed re-analysis or additional analysis of existing data, as well as text-based changes, it improved the clarity and completeness of the manuscript.

I do encourage the authors to expand their phenotyping assessments in future studies given that the interaction between structural disease, inflammation, and pain is complex, and our understanding of how the two interact and affect each other is evolving. There are multiple recent publications that show that a therapeutic or knock-out is protective against cartilage damage but doesn't alleviate pain, or vice versa. Thus, as a field, understanding which therapies target which pathological manifestations is an important next step to advance treatments. I also look forward to the follow-up studies on the MSC's role in ectopic cartilage.

---

## [Author Response]

The following is the authors’ response to the original reviews.

**Public Reviews:**

**Reviewer #1 (Public review):**
Summary:This work by Al-Jezani et al. focused on characterizing clonally derived MSC populations from the synovium of normal and osteoarthritis (OA) patients. This included characterizing the cell surface marker expression in situ (at time of isolation), as well as after in vitro expansion. The group also tried to correlate marker expression with trilineage differential potential. They also tested the ability of the different subpopulations for their efficacy in repairing cartilage in a rat model of OA. The main finding of the study is that CD47hi MSCs may have a greater capacity to repair cartilage than CD47lo MSCs, suggesting that CD47 may be a novel marker of human MSCs that have enhanced chondrogenic potential.Strengths:Studies on cell characterization of the different clonal populations isolated indicate that the MSC are heterogenous and traditional cell surface markers for MSCs do not accurately predict the differentiation potential of MSCs. While this has been previously established in the field of MSC therapy, the authors did attempt to characterize clones derived from single cells, as well as evaluate the marker profile at the time of isolation. While the outcome of heterogeneity is not surprising, the methods used to isolate and characterize the cells were well developed. The interesting finding of the study is the identification of CD47 as a potential MSC marker that could be related to chondrogenic potential. The authors suggest that MSCs with high CD47 repaired cartilage more effectively than MSC with low CD47 in a rat OA model.Weaknesses:While the identification of CD47 as a novel MSC marker could be important to the field of cell therapy and cartilage regeneration, there was a lack of robust data to support the correlation of CD47 expression to chondrogenesis. The authors indicated that the proteomics suggested that the MSC subtype expressed significantly more CD47 than the non-MSC subtype. However, it was difficult to appreciate where this was shown. It would be helpful to clearly identify where in the figure this is shown, especially since it is the key result of the study. The authors were able to isolate CD47hi and CD47 low cells. While this is exciting, it was unclear how many cells could be isolated and whether they needed to be expanded before being used in vivo. Additional details for the CD47 studies would have strengthened the paper. Furthermore, the CD47hi cells were not thoroughly characterized in vitro, particularly for in vitro chondrogenesis. More importantly, the in vivo study where the CD47hi and CD47lo MSCs were injected into a rat model of OA lacked experimental details regarding how many cells were injected and how they were labeled. No representative histology was presented and there did not seem to be a statistically significant difference between the OARSI score of the saline injected and MSC injected groups. The repair tissue was stained for Sox9 expression, which is an important marker of chondrogenesis but does not show production of cartilage. Expression of Collagen Type II would be needed to more robustly claim that CD47 is a marker of MSCs with enhanced repair potential.
**Reviewer #2 (Public review):**
Summary:This is a compelling study that systematically characterized and identified clonal MSC populations derived from normal and osteoarthritis human synovium. There is immense growth in the focus on synovial-derived progenitors in the context of both disease mechanisms and potential treatment approaches, and the authors sought to understand the regenerative potential of synovial-derived MSCs.Strengths:This study has multiple strengths. MSC cultures were established from an impressive number of human subjects, and rigorous cell surface protein analyses were conducted, at both pre-culture and post-culture timepoints. In vivo experiments using a rat DMM model showed beneficial therapeutic effects of MSCs vs non-MSCs, with compelling data demonstrating that only "real" MSC clones incorporate into cartilage repair tissue and express Prg4. Proteomics analysis was performed to characterize non-MSC vs MSC cultures, and high CD47 expression was identified as a marker for MSC. Injection of CD47-Hi vs CD47-Low cells in the same rat DMM model also demonstrated beneficial effects, albeit only based on histology. A major strength of these studies is the direct translational opportunity for novel MSC-based therapeutic interventions, with high potential for a "personalized medicine" approach.Weaknesses:Weaknesses of this study include the rather cursory assessment of the OA phenotype in the rat model, confined entirely to histology (i.e. no microCT, no pain/behavioral assessments, no molecular readouts). It is somewhat unclear how the authors converged on CD47 vs the other factors identified in the proteomics screen, and additional information is needed to understand whether true MSCs only engraft in articular cartilage or also in ectopic cartilage (in the context of osteophyte/chondrophyte formation). Some additional discussion and potential follow-up analyses focused on other cell surface markers recently described to identify synovial progenitors is also warranted. A conceptual weakness is the lack of discussion or consideration of the multiple recent studies demonstrating that DPP4+ PI16+ CD34+ stromal cells (i.e. the "universal fibroblasts") act as progenitors in all mesenchymal tissues, and their involvement in the joint is actively being investigated. Thus, it seems important to understand how the MSCs of the present study are related to these DPP4+ progenitors. Despite these areas for improvement, this is a strong paper with a high degree of rigor, and the results are compelling, timely, and important.Overall, the authors achieved their aims, and the results support not just the therapeutic value of clonally-isolated synovial MSCs but also the immense heterogeneity in stromal cell populations (containing true MSCs and non-MSCs) that must be investigated further. Of note, the authors employed the ISCT criteria to characterize MSCs, with mixed results in pre-culture and post-culture assessments. This work is likely to have a longterm impact on methodologies used to culture and study MSCs, in addition to advancing the field's knowledge about how synovial-derived progenitors contribute to cartilage repair in vivo.
**Recommendations for the authors:**

**Reviewer #1 (Recommendations for the authors):**
In all figures, it would be beneficial to report the sample number used for the data reported. It is difficult to appreciate the statistical analysis without that information.

Understood, the sample number and replicates have been added to each figure legend.

Please check that Table S7 is part of the manuscript. It could not be found.

It was added as an additional excel file since it was too large to fit in the word document.

Lines 377-379 (Figure 2E): the authors write that rats receiving MSCs had a significantly lower OARSI and Krenn score vs. rats injected with non-MSCs. However, none of the bars indicating statistical significance run between these two groups. Please verify the text and figure.

This has been corrected

The details surrounding the labeling of the cells with tdTomato were not presented in the methods.

This has been added to the methods

The fluorescent antibodies used should be listed and more details provided in the methods rather than a general statement that fluorescent antibodies were used.

Our apologies, the clones and companies have been added.

Additional information on the CD47 experiments (# cells, # animals) would have strengthened the study.

This has been added to the methods and figure legend.

**Reviewer #2 (Recommendations for the authors):**
My comments span minor corrections, requests for additional analyses, some suggestions for additional experiments, and requests for additional discussion of recent important studies.Introduction:The introduction is thorough and well-written. I recommend a brief discussion about the emerging evidence demonstrating that DPP4+ PI16+ CD34+ synovial cells, i.e. the "universal fibroblasts", act as stromal progenitors in development, homeostasis, and disease. Relevant osteoarthritis-related papers encompass human and mouse studies (PMIDs: 39375009, 38266107, 38477740, 36175067, 36414376).

This has been added.

Relatedly, as DPP4 is CD26 and therefore useful as a cell-surface antigen for flow cytometry, sorting, etc, it would be interesting to understand the relationship and similarities between the CD47-High cells identified in this study and the DPP4/PI16+ cells previously described. Do they overlap in phenotype/identity?

We have added a new flow cytometry figure for address this question.

Results:Note type-o on Line 311: "preformed" instead of "performed". Line 313 "prolife" instead of "profile"

Thank you for catching these.

The identified convergence of the cell surface marker profile of all normal and OA clones in culture is a highly intriguing result. Do the authors have stored aliquots of these cells to demonstrate whether this would also occur in soft substrate, i.e. low stiffness culture conditions? This could be done with standard dishes coated with bulk collagen or with commercially available low-stiffness dishes (1 kPa). This is relevant to multiple studies demonstrating the induction of a myofibroblast-like phenotype by stromal cells cultured on high-stiffness plastic or glass. This is also the experiment where assessment of DPP4/CD26 could be added, if possible.

While we agree it would be interesting to determine the mechanism by which the cells phenotypes converge, we would argue that it is outside of the scope of the current manuscript. We have instead added a sentence to the discussion.

Line 353 regarding the use of CD68 as a negative gate: can the authors pleasecomment on why they employed CD68 here and not CD45? While monocytes/macs/myeloid cells are the most abundant immune cells in synovium, CD45 would more comprehensively exclude all immune cells.

That is a fair point, and we really don’t have any reason to have picked CD68 over CD45. In our opinion either would be a fair negative marker to use based on the literature.

Fig 2, minor suggestion: consider adding "MSC vs non-MSC" on the experimental schematic to more comprehensively summarize the experiment.

This has been modified

Fig 2E should show all individual datapoints, not just bar graphs.

This has been modified

Fig 2: Given the significant reduction in Krenn score in DMM-MSC injected knees compared to DMM-saline knees, Fig 2 should also show representative images of the synovial phenotype to demonstrate which aspects of synovial pathology were mitigated. Was the effect related to lining hyperplasia, subsynovial infiltrate, fibrosis, etc? Similarly, can the authors narrate which aspects of the OARSI score drove the treatment effect (proteoglycans vs structure vs osteophytes, etc).

We have added a new sup figure breaking down the Krenn score as well as higher magnification images of representative synovium.

Fig 2: In the absence of microCT imaging, can the authors quantify subchondral bone morphometrics using multiple histological sections? The tibial subchondral bone in Fig 2D appears protected from sclerosis/thickening.

Unfortunately, this is beyond what are able to add to the manuscript.

The Fig 3 results are highly compelling and interesting. Congratulations.

Thank you very much.

Fig 4A: the cell highlighted in the high-mag zoom box in Fig 4A appears to be localized within the joint capsule or patellar tendon (it is unclear which anatomic region this image represents). The highly aligned nature of the tissue and cells along a fibrillar geometry indicates that this is not synovium. The interface between synovium and the tissue in question can be clearly observed in this image. Please choose an image more representative of synovium.

We completely agree with the reviewers assessment. However, it is the synovium that overlays this tissue (Fig 4A arrow). We are simply showing that there were very few MSCs that took up residence in the synovium or the adjacent tissues.

Fig 4C and F: please show individual data points.

This has been added

Fig 5D: I see DPP4 and ITGA5 were also hits in the proteomics analysis, which is intriguing. Besides my comments/suggestions regarding DPP4 above, please note this recent paper identifying a ITGA5+ synovial fibroblast subset that orchestrates pathological crosstalk with lymphocytes in RA, PMID: 39486872

Thank you for the information. We have added the reference in the results section.

Fig 5B-D: How did the authors converge on CD47 as the target for follow-up study? It does not appear to be a differentially-expressed protein based on the Volcano plot in Fig 5B, and it's unclear why it is a more important factor than any of the other proteins shown in the network diagram in Fig 5D, e.g. CTSL, ITGA5, DPP4. Can the authors add a quantitative plot supporting their statement "the MSC sub-type expressed significantly more CD47 than the non-MSCs" on Line 458?

We have re-written this line. It was incorrect to discuss amount of CD47. That was shown later with the flow analysis.

Fig 6D: Please show individual data points and also representative histology images to demonstrate the nature of the phenotypic effect.

This has been added.

Fig 6E-F: In what anatomic region are these images? Please add anatomic markers to clarify the location and allow the reader to interpret whether this is articular cartilage or ectopic cartilage

We have redone the figure to show the area as requested.

Relevant to this, do the authors observe this type of cellular engraftment in ectopic cartilage/osteophytes or only in articular cartilage? Understanding the contribution of these cells to the formation/remodeling of various cartilage types in the context of OA is a critical aspect of this line of investigation.

We didn’t see any contribution of these cells to ectopic cartilage formation and are actively working on a follow up study discussing this point specifically.

Discussion:Besides my comments regarding DPP4 and ITGA5 above, the authors may also consider discussing PMID: 37681409 (JCI Insight 2023), which demonstrates that adult Prg4+ progenitors derived from synovium contribute to articular cartilage repair in vivo.

We agree that there are numerous markers we could look at in future studies and that other people in the field are actively studying.